# Breastfeeding as a Strategic Driver for One Health: A Narrative Review

**DOI:** 10.3390/nu17233766

**Published:** 2025-11-30

**Authors:** Vanessa Machado, Simone Cecílio Hallak Regalo, Luciano Maia Alves Ferreira, Roberta Lopes de Castro Martinelli, Luciana Vitaliano Voi Trawitzki, Selma Siéssere, José João Mendes, João Botelho

**Affiliations:** 1School of Dentistry of Ribeirão Preto, University of São Paulo, Ribeirão Preto 14049-900, SP, Brazil; simone@forp.usp.br (S.C.H.R.); luvoi@fmrp.usp.br (L.V.V.T.); selmas@forp.usp.br (S.S.); 2Egas Moniz Center for Interdisciplinary Research (CiiEM), Egas Moniz School of Health and Science, 2829-511 Almada, Portugal; lucianomaia@egasmoniz.edu.pt (L.M.A.F.); jmendes@egasmoniz.edu.pt (J.J.M.); jbotelho@egasmoniz.edu.pt (J.B.); 3Lingual Frenulum Inspection Department, Hospital Santa Therezinha, Brotas 17380-000, SP, Brazil; robertalcm@gmail.com

**Keywords:** breastfeeding, One Health, human health, environmental health, animal health, economic, social

## Abstract

Breastfeeding is a renewable biological system that simultaneously advances human, environmental, and societal health. Human milk provides unparalleled nutrition and immunological protection, improving infant survival, neurodevelopment, and long-term metabolic outcomes, while reducing maternal risk of breast and ovarian cancer. However, and despite decades of evidence, only 48% of infants under six months are exclusively breastfed worldwide, and breastfeeding remains absent from most sustainability and One Health strategies. This narrative review synthesizes evidence demonstrating that breastfeeding functions as a low-carbon, zero-waste food system that avoids greenhouse gas emissions, land conversion, water consumption, and biodiversity loss linked to commercial milk formula production. At the societal level, breastfeeding reduces health-system costs, strengthens emergency resilience when supply chains fail, and generates long-term economic returns. By integrating evidence across human health, environmental impact and social determinants, this review positions breastfeeding as a strategic One Health intervention and a high-value investment for achieving multiple Sustainable Development Goals. Strengthening policy support—including protection against formula marketing, workplace accommodations, and expansion of baby-friendly systems—is essential to unlock breastfeeding’s potential for planetary and public health.

## 1. Introduction

Breastfeeding is recognized as a biological norm and a fundamental component of public health [1], providing essential nutrition, immune protection, and long-term health benefits for both infants and mothers [2,3]. Despite decades of substantial evidence, breastfeeding is frequently perceived narrowly as an individual maternal-infant behavior rather than as a strategic intervention with broad systemic impact [4]. This limited perspective has obscured its potential to act as a catalyst for the One Health approach, which acknowledges the interdependence of human, animal, and environmental health [5,6,7].

At the global level, breastfeeding practices have improved but remain inadequate to meet international targets [3,5]. Progress is uneven across and within countries, influenced by social determinants such as labor protection and health system support. Beyond health, infant feeding choices have implications for planetary sustainability. While human milk is naturally renewable and low-carbon, formula production necessitates intensive livestock farming, water and energy use, and generates waste and greenhouse gas emissions [8,9].

Although these intersections with health, equity, and environment are increasingly recognized, breastfeeding is seldom explicitly integrated into One Health or sustainability frameworks [10]. Its contributions to resilient food systems, climate mitigation, and biodiversity protection remain largely invisible in policy agendas and measurement tools.

This review integrates breastfeeding into the One Health and sustainability discourse by synthesizing evidence across human, animal, and environmental dimensions (Figure 1). It explores how breastfeeding contributes to global health promotion, environmental protection, and social equity, while identifying policy gaps that hinder its recognition as a systemic intervention. By reframing breastfeeding as a cornerstone of sustainable development, this paper promotes viewing it as a collective societal responsibility with significant implications for global resilience.

## 2. Prevalence and Trends of Breastfeeding Globally

Over the past two decades, global breastfeeding rates have shown a steady improvement, though progress remains uneven and insufficient to meet international targets. Recent monitoring report show that 48% of infants aged under 6 months are exclusively breastfed, representing an increase of more than 10 percentage points since 2012 and nearing the interim 50% target for 2025, but still far from the 70% goal for 2030 [1]. Other key indicators remain below global ambitions: only 46% of newborns initiate breastfeeding within the first hour of life, 72% of children are still breastfed at one year, and 46% continue breastfeeding at two years, short of the recommended thresholds of 70%, 80%, and 60%, respectively [1]. These global averages conceal marked disparities. Between 2017 and 2022, 75 of 106 assessed countries increased exclusive breastfeeding rates, with 23 achieving gains of than 10 percentage points, whereas 29 countries experienced declines [1], often linked to rapid socioeconomic transition, urbanization, and the pervasive marketing of commercial milk formulas [2,3,4].

Low- and middle-income countries (LMICs) have driven much of the overall progress, with significant increases in exclusive breastfeeding through 6 months and improved early initiation [5]. However, harmful practices such as pre-lacteal feeding continue to undermine breastfeeding, affecting approximately one in three newborns and undermining both early initiation and breastfeeding duration [6]. Global burden of disease analyses confirm that the health cost of suboptimal breastfeeding, though decreasing, remains considerable: between 1990 and 2019, years of life lost attributable to non-exclusive or discontinued breastfeeding in children under five declined by approximately 70% worldwide, yet the greatest absolute burden persists in sub-Saharan Africa and the Middle East/North Africa [7]. These reflect improvements in child survival but continued exposure to non-optimal feeding practices, including in high-income contexts.

High-income countries show encouraging but incomplete progress. In the United States, breastfeeding initiation increased from about 68% in 1999–2000 to nearly 88% in 2017–2018 [11], but only 24.9% were exclusively breastfed at 6 months of age [12]. Despite these gains, persistent disparities by socioeconomic status and race/ethnicity reveal structural barriers such as limited paid parental leave, inconsistent workplace support, and inequitable access to lactation counseling [13]. Socioeconomic status and racial or ethnic identity frequently serve as structural impediments to breastfeeding. Women from lower-income backgrounds encounter challenges such as shorter maternity leave, unstable employment, and restricted access to lactation-friendly environments, all of which contribute to the premature cessation of breastfeeding [3,5]. In numerous high-income nations, including the United States, breastfeeding rates among Black mothers remain significantly lower than those among White or Hispanic mothers [14]. This disparity reflects the cumulative impact of systemic inequities, cultural marginalization, and unequal healthcare support [14].

In Europe, the situation is more heterogeneous: the World Health Organization (WHO) European Region exhibits some of the lowest rates of exclusive breastfeeding at 6 months of age globally, with a median of approximately 23% as reported in studies, and significant variation between countries [15,16]. For instance, data indicate that exclusive breastfeeding under 6 months is as low as 1% in certain countries such as Greece, Finland, and the UK, while reaching approximately 49% in Slovakia [15]. More recently, in England, the rate of breastfeeding at 6–8 weeks increased to 52.7% in 2023–2024 [10]; however, substantial regional and socioeconomic disparities persist. These findings highlight that although high-income countries have made progress, many remain far from achieving the global recommendations for early initiation and sustained breastfeeding up to two years.

The considerable variation in breastfeeding prevalence across different countries is indicative of disparities not only in health system capacity and policy support but also in social norms, parental attitudes, and cultural perceptions regarding breastfeeding. Evidence has demonstrated that maternal confidence, perceived social approval, and exposure to marketing of breast-milk substitutes significantly influence both the initiation and duration of breastfeeding [17,18,19]. In numerous contexts, early return to work, insufficient partner and family support, and discomfort with public breastfeeding serve as substantial deterrents [3,20,21], whereas paternal involvement and positive attitudes towards breastfeeding are correlated with extended duration and exclusivity [3,18]. Additionally, cultural ideals of modernity and body image frequently discourage breastfeeding in high-income settings [4,17], while in certain low-income contexts, social stigma and misinformation continue to undermine optimal feeding practices [3,6]. These behavioral and sociocultural dynamics illustrate that breastfeeding practices are not determined solely by individual choice but by structural and cultural environments that either facilitate or constrain parental agency [3,17,20]. In both developed and developing countries, these factors intersect with economic insecurity, limited maternity protection, and insufficient access to trained lactation support, creating a structural environment that hinders breastfeeding continuation [3,5]. Addressing such barriers requires comprehensive policies that integrate labor rights, health system support, and community-based interventions.

Overall, exclusive breastfeeding globally is on an upward trajectory and may meet the 2025 target, but early initiation and continued breastfeeding to two years remain below WHO recommendations worldwide. Although the disease burden associated with artificial feeding is declining, it continues to generate preventable mortality and developmental loss, disproportionately affecting vulnerable regions [7]. Achieving the 2030 global nutrition and child survival targets will require accelerated implementation of evidence-based measures—full enforcement of the International Code of Marketing of Breast-milk Substitutes, universal adoption of the Baby-Friendly Hospital Initiative, comprehensive maternity protection, and accessible skilled lactation support—to convert partial advances into sustained, equitable global impact [3,17].

## 3. Breastmilk and Human Health

Breastfeeding extends beyond being a mere source of infant nutrition and represents a multifaceted biological and social process with significant, enduring health implications for infants, mothers and societies. Human milk is a dynamic and living biological fluid, comprising not only macronutrients but also immune cells, antibodies, bioactive peptides, human milk oligosaccharides (HMOs), hormones, and commensal bacteria that influence gut microbiome colonization and immune maturation [17,22,23,24,25]. Exclusive breastfeeding during the first six months of life offers protection to infants against acute infections, including diarrheal disease, pneumonia, and otitis media, while also reducing all-cause infant mortality [5,26]. The long-term benefits of breastfeeding extend to neurodevelopment, with consistent associations observed between breastfeeding and higher intelligence quotient (IQ) scores as well as improved educational attainment [27,28,29]. Notably, breastfeeding provides protection against later-life overweight or obesity [30], and type 2 diabetes through pathways involving metabolic and appetite regulation [31].

In mothers, substantial epidemiological evidence indicates that breastfeeding provides protective effects against several major non-communicable diseases (NCDs) in women. Increased cumulative duration of breastfeeding is associated with reduced risk of breast cancer [32,33] and ovarian cancer [34,35,36], likely mediated by lactational amenorrhoea, hormonal modulation, and reduced lifetime exposure to estrogen [17,29]. Breastfeeding also improves metabolic regulation, contributing to a lower incidence of type 2 diabetes and hypertension later in life [3,37]. Additionally, lactational amenorrhoea supports natural birth spacing, which enhances maternal health and reduces risks associated with short interpregnancy intervals.

Breastfeeding plays a nuanced role in early oral health. Evidence indicates that exclusive breastfeeding up to 6 months and breastfeeding within the first year of life may protect against malocclusion and support healthy craniofacial development [38,39,40,41,42]. In another point of view, recent systematic reviews and meta-analyses show that prolonged breastfeeding, particularly beyond 12–18 months and with frequent nocturnal feeds, is associated with an increased risk of early childhood caries (ECC) [43,44,45,46]. For instance, Shrestha et al. [45] found that breastfeeding ≥ 12 months increased ECC odds compared with breastfeeding for shorter durations, and Carrillo-Díaz et al. [47] reported worse oral-health-related quality of life and higher caries experience among children engaging in prolonged night-time breastfeeding and co-sleeping [47]. Despite these findings, the causal nature of this association remains uncertain. Most evidence comes from observational studies, often heterogeneous in exposure definitions (exclusive vs. any breastfeeding, duration thresholds), and do not measure post-feeding oral hygiene behaviors, such as whether brushing occurs after nocturnal breastfeeding or parental awareness of oral-health guidance. Thus, the increased ECC risk associated with prolonged breastfeeding may reflect not only the biological effect of prolonged exposure to milk sugars when salivary flow is low during sleep, but also behavioral factors such as inadequate oral hygiene, high sugar intake, or delayed dental visits. To determine whether breastfeeding itself is cariogenic or whether risk stems from modifiable parental behaviors, future research requires well-designed longitudinal intervention studies that track feeding patterns, oral-hygiene practices, diet, and fluoride exposure from infancy through tooth eruption.

## 4. Breastmilk Impact on Environmental & Animal Health

Breastfeeding represents not only the biological standard for infant nutrition but also a practice that is sustainable and environmentally friendly. In contrast to commercial milk formula (CMF), breastmilk is produced and consumed directly, eliminating the need for industrial processing, transportation, packaging, or energy for preparation. This inherent efficiency positions breastfeeding as a low-carbon, waste-free food system that aligns with One Health and sustainability objectives.

The production of CMF is notably resource-intensive, with dairy farming and industrial processing contributing significantly to greenhouse gas emissions and consuming substantial water and energy resources [8,9]. Additionally, the processes of sterilization and preparation further exacerbate environmental costs [48]. Conversely, breastfeeding has a minimal carbon footprint, requires no external water beyond maternal hydration, and circumvents the energy demands associated with formula production and household preparation [8,37]. It is also virtually zero-waste, as it does not generate tins, plastics, bottles, teats, or shipping materials, thereby reducing solid waste and marine pollution [9].

The production of formula also contributes to land conversion and biodiversity loss, particularly through dairy and soy supply chains, and raises concerns regarding animal welfare due to intensive livestock farming [49]. Breastfeeding circumvents these extractive systems, offering a self-renewing, species-specific food source that does not compete for arable land or disrupt ecosystems. Furthermore, human milk donation and milk banks can amplify these environmental advantages. Donor milk provides safe, biologically appropriate nutrition for infants unable to receive their own mother’s milk, thereby decreasing the demand for formula in neonatal intensive care and other vulnerable settings. Although the processing and storage of donor milk require some resources, the overall ecological footprint remains significantly lower than that of the industrial production, packaging, and distribution of CMF [50].

Global health agencies increasingly acknowledge human milk—whether from the mother or donor networks—as integral to sustainable food systems that bolster local resilience and reduce reliance on environmentally detrimental supply chains [37]. Positioning breastfeeding and donor milk provision as sustainable feeding practices shifts the discourse from individual maternal responsibility to collective health and ecological preservation. Enhancing maternity protections, enforcing the International Code of Marketing of Breast-milk Substitutes, expanding access to regulated human milk banks, and investing in baby-friendly health systems are crucial measures to diminish reliance on formula and its environmental impact [48,51].

## 5. One Health & Societal Dimensions

Breastfeeding transcends individual choice, functioning as a societal resilience mechanism that enhances health, promotes equity, and supports sustainable systems. From a One Health perspective—acknowledging the interconnections between human health, social structures, and the environment—it emerges as a crucial public good rather than a private responsibility [3,17].

Because breastmilk is sterile, safe and immediately available, breastfeeding acts as a lifeline during emergencies. In crises such as natural disasters, conflict, or supply chain disruptions, it ensures infant survival when safe water and formula preparation are not guaranteed and alleviates pressure on fragile health services [52]. Emergency management agencies now recognize breastfeeding support as part of disaster preparedness and food security policies [1]. Formula-dependent infants are highly vulnerable when electricity, transportation, or sterile water become unavailable, increasing risks of dehydration, diarrheal disease and mortality [52].

Breastfeeding capacity is deeply shaped by social determinants of health. Adequate paid maternity leave, workplace protections, and access to lactation rooms substantially extend breastfeeding duration [17,18,19,20,21]. Conversely, precarious employment, lack of job protection, and absence of skilled lactation care led women to cease breastfeeding prematurely. These inequities show that breastfeeding is not only about maternal “choice”, but about structural conditions that either enable or obstruct it.

Breastfeeding also embodies cultural identity and intergenerational knowledge, yet these traditions are increasingly disrupted by commercial marketing that normalizes formula use and undermines maternal confidence [2,4]. Protecting families from misleading advertising and bolstering community support are vital to sustaining breastfeeding-friendly cultures.

Economically, breastfeeding represents a high-return investment. Families save on feeding costs, and health systems avoid expenditures related to infections, malnutrition, and chronic diseases. Globally, suboptimal breastfeeding leads to US $341 billion in economic losses annually through lost productivity and increased treatment costs [53]. When viewed through the One Health lens, breastfeeding is not merely a maternal behavior; it constitutes a collective infrastructure for health resilience, social justice, and sustainable development [17].

## 6. Policy and Practice Implications

Positioning breastfeeding as a One Health intervention requires going beyond the traditional maternal-child health programs to integrate breastfeeding comprehensively across health, environmental, and social policies. Major global frameworks have already provided impetus: WHO and Food and Agriculture Organization (FAO) advocate for food systems that are safe, sustainable and resource-efficient, and planetary health agendas increasingly acknowledge human milk as a renewable, low-impact food resource [54]. Aligning breastfeeding promotion with these agendas amplifies its relevance to climate mitigation, food system transformation and sustainability planning [2].

The Global Breastfeeding Collective, spearheaded by the WHO and the United Nations International Children’s Emergency Fund (UNICEF), delineates seven policy priorities that serve as a strategic framework for governments and stakeholders to foster environments conducive to breastfeeding [3,5,55]. These priorities encompass increasing financial investment to safeguard, promote, and support breastfeeding; adopting and enforcing the International Code of Marketing of Breast-milk Substitutes; enacting policies for paid family leave and workplace breastfeeding; implementing the Baby-Friendly Hospital Initiative; enhancing access to skilled breastfeeding counseling; fortifying community support systems; and monitoring progress through robust data collection [3,5,55]. Integrating these priorities into national and local health and sustainability agendas underscores breastfeeding as a societal investment in equity, resilience, and planetary well-being.

Within health systems, creating protective and enabling environments is essential. Expanding and updating the Baby-Friendly Hospital Initiative improves breastfeeding initiation and continuation particularly when combined with postnatal support and continuity of care [1,17,56]. Nevertheless, the Baby-Friendly Hospital Initiative has achieved full implementation in fewer than 10% of maternity facilities worldwide, underscoring disparities in national adoption, monitoring, and financing [56].

Enhancing the training of healthcare personnel and fostering community engagement are crucial for the successful implementation of breastfeeding-supportive frameworks. Yet, implementation remains inconsistent and underfunded. Training of the health workforce also exhibits marked variability, with many professionals reporting insufficient practical skills and limited integration of breastfeeding into pre-service and continuing education curricula. Strengthening these dimensions through standardized training and robust monitoring, combined with community-level interventions such as peer counseling networks, mother-to-mother support groups, and home visits, can substantially increase breastfeeding initiation and continuation rates [5,57]. Examples such as the ‘Kangaroo Mother Care’ approach and the ‘Alive & Thrive’ initiative demonstrate how strong linkage between maternity services and community-based follow-up can sustain breastfeeding practices and bolster maternal confidence [58,59]. Addressing this systemic gaps through integrated, well-funded, and multisectoral strategies are essential for realizing the full potential of breastfeeding as a cornerstone of the One Health framework.

Integrating breastfeeding counseling into primary care, digital health solutions, and public health communication strengthens maternal confidence and counters commercial milk formula marketing [2]. Evidence shows that restricting formula marketing and enforcing the International Code of Marketing of Breast-milk Substitutes significantly increases exclusive breastfeeding rates [51]. Updates should address gaps identified since its global launch, such as insufficient staff training, weak monitoring and accreditation systems, and inequitable implementation in low-resource settings. Revising this initiative also may consider new determinants of breastfeeding practices, including the influence of digital marketing of breast-milk substitutes and the need for alignment with current WHO and UNICEF guidance on maternity protection, gender equity, and sustainability goals.

Environmental and climate policies can also drive meaningful change. Sustainable food system frameworks and national climate action plans rarely incorporate breastfeeding, despite its demonstrated contribution to reduction in greenhouse gas emissions, waste and water consumption when compared to commercial milk formula [8,51]. Recognizing breastfeeding and donor human milk systems as low-carbon, circular, waste-free food strategies supports environmental conservation goals [49].

Finally, interdisciplinary collaboration is key. Health professionals, environmental scientists, economists, food system experts, and policymakers must work collectively to embed breastfeeding in climate strategies, nutritional policies, labor protections, and gender equity agendas [17,53]. Such a cross-sector approach reframes breastfeeding not merely as a maternal behavior, but as a societal investment that simultaneously advances health, sustainability, equity and resilience.

## 7. Challenges and Knowledge Gaps

Despite the increasing recognition of breastfeeding as a sustainable and One Health intervention, significant challenges and evidence gaps hinder its comprehensive integration into health, environmental, and policy agendas.

A central obstacle is the fragmented framing of breastfeeding. Although global strategies from the WHO, FAO, and planetary health initiatives increasingly advocate for sustainable, safe food systems, breastfeeding is still rarely positioned within these frameworks as a renewable, low-impact food source [17,54]. Without stronger cross-sectoral language and coordinated indicators, its contributions to climate mitigation, biodiversity protection, circular economy and food sovereignty remain overlooked [51].

Within health systems, the scaling of effective breastfeeding support is inconsistent. While the Baby-Friendly Hospital Initiative improves breastfeeding initiation and duration, implementation remains limited, training is heterogeneous, and continuity of care into the community is frequently underfunded [1,17]. Simultaneously, aggressive CMF marketing undermines confidence and exploits regulatory loopholes, directly countering health system efforts [2,4].

The environmental dimension of infant feeding remains under-researched. Although several life-cycle assessments quantify the greenhouse gas emissions and resource demand of formula production [8,49], data on the environmental savings from breastfeeding and donor human milk systems are scarce. Few studies model the integration of breastfeeding into national carbon budgets or climate action plans, limiting its visibility in environmental policy [51].

Persistent labor and social protection gaps impede equitable breastfeeding. Many countries fail to provide adequate paid parental leave, workplace accommodations for lactation, or enforcement of the International Code of Marketing of Breast-milk Substitutes [20,21,60]. These structural inequities disproportionately affect women with precarious employment and low income, contributing to early weaning and perpetuating health disparities.

Finally, interdisciplinary collaboration is limited. Research communities focusing on maternal–child health, sustainability, climate change, and food systems often work in silos. Bridging these gaps requires shared metrics, cross-sector research, and joint advocacy between clinicians, environmental scientists, economists, nutritionists, and policymakers [2].

## 8. Conclusions

Breastfeeding is a high-value intervention that simultaneously benefits human health, strengthens social resilience, and reduces environmental impact. By providing optimal nutrition and immune protection for infants and lowering maternal risk for major NCD, breastfeeding improves population health while reducing healthcare costs. Unlike commercial milk formula, breastfeeding is naturally renewable, generates no waste, and avoids resource-intensive production systems, positioning it as an environmentally sustainable food source within a One Health framework.

Despite this evidence, breastfeeding remains insufficiently integrated into environmental, food system, and climate policies. Structural barriers—including inadequate maternity protection, limited access to skilled support, and pervasive formula marketing—continue to hinder breastfeeding practices globally.

Recognizing breastfeeding as essential public health infrastructure rather than an individual responsibility is crucial. Aligning health, labor, and environmental policies; enforcing the International Code of Marketing of Breast-milk Substitutes; and scaling Baby-Friendly and community support initiatives are key to unlocking breastfeeding’s full benefits. Elevating breastfeeding within One Health and sustainability agendas offers a pragmatic, low-cost strategy to advance health equity, strengthen social and emergency resilience, and reduce environmental harm.

## Figures and Tables

**Figure 1 nutrients-17-03766-f001:**
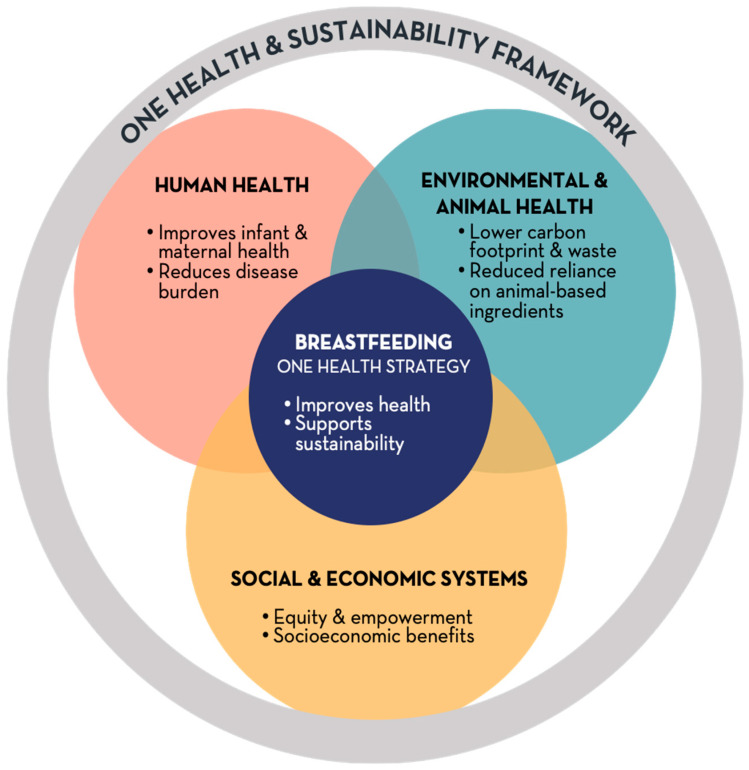
Conceptual model of breastfeeding as a One Health strategy. Breastfeeding sits at the intersection of human, environmental and societal systems. It improves infant and maternal health, reduces the burden on health services, and generates social and economic benefits while minimizing environmental impact and reliance on animal-based production. Integrating breastfeeding into One Health and sustainability frameworks strengthens resilience, equity and planetary health.

## Data Availability

Not applicable.

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
