# Peer review of "Breastfeeding as a Strategic Driver for One Health: A Narrative Review"

_nutrients, 2025, doi:10.3390/nu17233766_

Round 1
Reviewer 1 Report
Comments and Suggestions for Authors
The Authors discuss a relevant topic: breastfeeding as a strategic determinant within the One Health framework to enhance resilience, equity, and planetary health. They highlight that, although breastfeeding is acknowledged as a fundamental component of public health and sustainability, it is seldom made explicit within One Health or sustainability policies and frameworks. This review provides a rationale for integrating breastfeeding into these discourses by pointing it as a collective social responsibility rather than an individual maternal-infant behavior. The Authors synthesize evidence on the benefits of breastfeeding for the health of both mother-child dyad and the Planet, and the societal well-being as well, while identifying challenges and gaps that prevent breastfeeding from fully meeting recommendations and being adequately represented in sustainability and One Health strategies. The paper is interesting and well written. Nonetheless, it would benefit from some improvements. Some points should be slightly expanded, further detailed, or improved by concrete examples. Accordingly, some revisions are recommended to further enhance the quality of the manuscript.
- Lines 96–99: Please, consider adding evidence-based examples illustrating how socioeconomic status and/or race and ethnicity act as barriers to breastfeeding
- Starting from the “Policy and Practice Implications” section, the numbering needs to be revised, as the same enumeration, namely 5, repeats up to the “Conclusion” section
- Please, consider to position the “Policy and Practice Implications” section after “Challenges and Gaps” section. It is likely more logical, as policy considerations may constitute the appropriate place to address opportunities to tackle the identified challenges. Within the section “Policy and Practice Implications”, the Authors should consider mentioning the Global Breastfeeding Collective policy priorities by WHO and UNICEF. Furthermore, the Authors should discussion more explicitly the key role of improved training of health personnel and enhanced community engagement. Including concrete examples of how strengthening links between health facilities and communities, as well as of family- and community-based interventions/initiatives improving antenatal and postnatal breastfeeding support would be helpful. This is particularly relevant given that heterogeneous training and limited continuity of care at the community level are identified as barriers (as the authors correctly state at line 279)
- Line 247: Please, elaborate on what “updating the Baby-Friendly Hospital Initiative” entails, specifying which components require revision and why
- Lines 279–280: While the statement that “implementation remains limited, training is heterogeneous, and continuity of care into the community is frequently underfunded” is valid, further explanation is needed. Please, clarify the evidence on which these conclusions are based.
- References throughout the text. Line 45: Reference 8 is described as addressing the global level; however, it deals only with children in the United States. Please, replace it with a more appropriate reference. Lines 50 and 54: References 9–13 relate to breastfeeding rates, not to sustainability. Please, substitute them with citations that address sustainability dimensions. Line 415: Please, update the reference to: “Cenzato N, Berti C, Cazzaniga F, Di Iasio G, Scolaro A, Maspero C. Influence of the type of breastfeeding as a risk or protective factor for the onset of malocclusions: a systematic review. Eur J Paediatr Dent. 2023;24(4):329-333. doi: 10.23804/ejpd.2023.2015”
- Line 316: The Authors correctly state that “community support initiatives are key to unlocking breastfeeding’s full benefits”. However, no specific examples are provided throughout the text. Please, include some examples of community support initiatives in the “Policy and Practice Implications” section, as mentioned above
Author Response
Thank you for giving us the opportunity to submit the revised draft of the manuscript “Breastfeeding as a strategic driver for One Health: Narrative Review” for publication in Nutrients in Section Pediatric Nutrition (ID number nutrients-3990346).
We appreciate the time and effort dedicated to providing feedback on our manuscript and are grateful for their insightful comments.
We have assessed and addressed all commentaries provided by all referees. These changes have been highlighted in the revised manuscript. Please see below, our point-by-point responses to the reviewer comments and concerns.
The pdf of the manuscript has all of the comments highlighted.
Reviewer #1
The Authors discuss a relevant topic: breastfeeding as a strategic determinant within the One Health framework to enhance resilience, equity, and planetary health. They highlight that, although breastfeeding is acknowledged as a fundamental component of public health and sustainability, it is seldom made explicit within One Health or sustainability policies and frameworks. This review provides a rationale for integrating breastfeeding into these discourses by pointing it as a collective social responsibility rather than an individual maternal-infant behavior. The Authors synthesize evidence on the benefits of breastfeeding for the health of both mother-child dyad and the Planet, and the societal well-being as well, while identifying challenges and gaps that prevent breastfeeding from fully meeting recommendations and being adequately represented in sustainability and One Health strategies. The paper is interesting and well written. Nonetheless, it would benefit from some improvements. Some points should be slightly expanded, further detailed, or improved by concrete examples. Accordingly, some revisions are recommended to further enhance the quality of the manuscript.
Our answer: We sincerely thank Reviewer #1 for the thoughtful and constructive comments. We appreciate the positive assessment of our manuscript and the recognition of its relevance and clarity. We have carefully considered all suggestions and revised the text accordingly to improve depth, clarity, and contextualization. Specifically, we have expanded sections where additional detail and concrete examples were warranted, ensuring that the discussion more effectively illustrates the integration of breastfeeding within the One Health and sustainability frameworks. Below, we provide a point-by-point response addressing each of the reviewer’s comments and outlining the corresponding changes made in the revised version.
Lines 96–99: Please, consider adding evidence-based examples illustrating how socioeconomic status and/or race and ethnicity act as barriers to breastfeeding
Our answer: We thank the reviewer for this valuable suggestion. We have revised the text to include concrete, evidence-based examples demonstrating how socioeconomic status and race/ethnicity influence breastfeeding practices and outcomes. The revised text now reads as follows: “Socioeconomic status and racial or ethnic identity frequently serve as structural impe-diments to breastfeeding. Women from lower-income backgrounds encounter challenges such as shorter maternity leave, unstable employment, and restricted access to lactation-friendly environments, all of which contribute to the premature cessation of breas-tfeeding (Victora et al., 2016; Rollins et al., 2023). In numerous high-income nations, including the United States, breastfeeding rates among Black mothers remain significantly lower than those among White or Hispanic mothers. This disparity reflects the cumulative impact of systemic inequities, cultural marginalization, and unequal healthcare support (Anstey et al., 2017).”
Starting from the “Policy and Practice Implications” section, the numbering needs to be revised, as the same enumeration, namely 5, repeats up to the “Conclusion” section
Our answer: We apologize for this typo and thank the carefulness. We have revised enumeration that now reads as follows in the revised manuscript: 6.Policy and Practice Implications; 7. Challenges and Knowledge Gaps; and 8. Conclusions.
Please, consider to position the “Policy and Practice Implications” section after “Challenges and Gaps” section. It is likely more logical, as policy considerations may constitute the appropriate place to address opportunities to tackle the identified challenges. Within the section “Policy and Practice Implications”, the Authors should consider mentioning the Global Breastfeeding Collective policy priorities by WHO and UNICEF. Furthermore, the Authors should discussion more explicitly the key role of improved training of health personnel and enhanced community engagement. Including concrete examples of how strengthening links between health facilities and communities, as well as of family- and community-based interventions/initiatives improving antenatal and postnatal breastfeeding support would be helpful. This is particularly relevant given that heterogeneous training and limited continuity of care at the community level are identified as barriers (as the authors correctly state at line 279)
Our answer: We are thankful for this suggestion. Accordingly, we have positioned the “Policy and Practice Implications” section after “Challenges and Gaps” section. In addition, we have mentioned the Global Breastfeeding Collective policy priorities by WHO and UNICEF in the second new paragraph. And subsequently discussed the key role of improved training of health personnel and enhanced community engagement with concrete examples.
Line 247: Please, elaborate on what “updating the Baby-Friendly Hospital Initiative” entails, specifying which components require revision and why
Our answer: We appreciate the reviewer’s insightful comment. We have expanded this section to clarify what is meant by “updating the Baby-Friendly Hospital Initiative” and to specify which components require revision and why. In the revised version, we explain that updates to this initiative should reflect the need for broader system integration. Linking maternity and newborn care to community and primary care services while addressing emerging challenges such as digital marketing of breast-milk substitutes, workforce training gaps, and equitable implementation across diverse socioeconomic and cultural contexts. We also emphasize that reinforcing post-discharge follow-up, community support, and adherence to the International Code of Marketing of Breast-milk Substitutes are critical to ensure the Initiative’s continued effectiveness and sustainability.
The new added text reads as follows: “Updates should address gaps identified since its global launch, such as insufficient staff training, weak monitoring and accreditation systems, and inequitable implementation in low-resource settings. Revising this initiative also may consider new determinants of breastfeeding practices, including the influence of digital marketing of breast-milk substitutes and the need for alignment with current WHO and UNICEF guidance on maternity protection, gender equity, and sustainability goals.”
Lines 279–280: While the statement that “implementation remains limited, training is heterogeneous, and continuity of care into the community is frequently underfunded” is valid, further explanation is needed. Please, clarify the evidence on which these conclusions are based.
Our answer: We thank the reviewer for this pertinent observation. We have now expanded the section to clarify the empirical evidence underpinning these conclusions. Specifically, we refer to global assessments showing that despite the proven effectiveness of breastfeeding initiatives, their implementation coverage remains limited—fewer than 10% of infants worldwide are born in facilities fully compliant with the Baby-Friendly Hospital Initiative (BFHI) criteria (WHO & UNICEF, 2018). Moreover, heterogeneity in health personnel training has been reported across regions, where gaps in pre-service and in-service breastfeeding education persist (Rollins et al., Lancet, 2023; Pérez-Escamilla et al., J Hum Lact, 2018). Finally, continuity of care and community-based support are frequently underfunded or fragmented, limiting post-discharge follow-up and the effectiveness of health–community linkages (Victora et al., Lancet, 2016; Lumbiganon et al., Cochrane Database Syst Rev, 2016). The revised text below integrates this evidence and provides explicit reference to the sources.
References throughout the text. Line 45: Reference 8 is described as addressing the global level; however, it deals only with children in the United States. Please, replace it with a more appropriate reference. Lines 50 and 54: References 9–13 relate to breastfeeding rates, not to sustainability. Please, substitute them with citations that address sustainability dimensions. Line 415: Please, update the reference to: “Cenzato N, Berti C, Cazzaniga F, Di Iasio G, Scolaro A, Maspero C. Influence of the type of breastfeeding as a risk or protective factor for the onset of malocclusions: a systematic review. Eur J Paediatr Dent. 2023;24(4):329-333. doi: 10.23804/ejpd.2023.2015”
Our answer: We thank the reviewer for the careful attention to the accuracy and relevance of the references. We have implemented all the suggested changes accordingly.
Line 316: The Authors correctly state that “community support initiatives are key to unlocking breastfeeding’s full benefits”. However, no specific examples are provided throughout the text. Please, include some examples of community support initiatives in the “Policy and Practice Implications” section, as mentioned above
Our answer: We appreciate this helpful suggestion. As also addressed in response to the previous comment (Lines 279–280), we have expanded the “Policy and Practice Implications” section to include concrete examples of community support initiatives.
Reviewer 2 Report
Comments and Suggestions for Authors
Dear Authors,
I consider the topic of breastfeeding to be timely and important, but in my opinion, the article requires improvement.
Firstly, I'm not sure if the literature is cited correctly – in Chapter 1, the authors cite references [1-13}, and then in Chapter 2, starting from reference number 1? – Is this correct, or is this a mistake? (It seems like an error).
If this is a scientific narrative review article, a more comprehensive approach would be useful – for example:
- Please provide more examples of the prevalence of breastfeeding in various countries (more countries to provide an overview).
- Breastfeeding is at varying levels – why is this? What are the opinions/attitudes of mothers (and fathers) towards breastfeeding?
- What contributes to such low breastfeeding rates? What are the barriers to breastfeeding in both developing and developed countries?
I believe that the above comments will enrich the topic addressed by the authors, and obtaining answers to these questions will help build strategies to raise awareness of the importance of breastfeeding.
Author Response
Thank you for giving us the opportunity to submit the revised draft of the manuscript “Breastfeeding as a strategic driver for One Health: Narrative Review” for publication in Nutrients in Section Pediatric Nutrition (ID number nutrients-3990346).
We appreciate the time and effort dedicated to providing feedback on our manuscript and are grateful for their insightful comments.
We have assessed and addressed all commentaries provided by all referees. These changes have been highlighted in the revised manuscript. Please see below, our point-by-point responses to the reviewer comments and concerns.
The pdf of the manuscript has all of the comments highlighted.
Reviewer #2
Dear Authors,
I consider the topic of breastfeeding to be timely and important, but in my opinion, the article requires improvement.
Our answer: We sincerely thank Reviewer #2 for the thoughtful and constructive comments. We appreciate the positive assessment of our manuscript and the recognition of its relevance. We have carefully considered all suggestions and revised the text accordingly to improve depth, clarity, and contextualization. Below, we provide a point-by-point response addressing each of the reviewer’s comments and outlining the corresponding changes made in the revised version.
Firstly, I'm not sure if the literature is cited correctly – in Chapter 1, the authors cite references [1-13}, and then in Chapter 2, starting from reference number 1? – Is this correct, or is this a mistake? (It seems like an error).
Our answer: We thank the reviewer for noticing this detail. The reference numbering has been carefully checked and corrected to ensure consistency throughout the manuscript.
If this is a scientific narrative review article, a more comprehensive approach would be useful – for example:
- Please provide more examples of the prevalence of breastfeeding in various countries (more countries to provide an overview).
Our answer: We appreciate the reviewer’s thoughtful suggestion. We respectfully note that the section “Prevalence and Trends of Breastfeeding Globally” was designed to provide a concise yet representative overview consistent with the scope of a narrative review. The revised text includes global indicators and progress toward the 2025 and 2030 targets, complemented by regional and national examples that reflect diverse socioeconomic and geographic contexts. Specifically, we present data for:
- Global trends, including exclusive, continued, and early initiation of breastfeeding and their evolution since 2012;
- Regional patterns, covering low- and middle-income countries, sub-Saharan Africa, and the Middle East/North Africa, highlighting both progress and persistent challenges;
- Country-level examples, including the United States (national trends and racial/ethnic disparities), the United Kingdom (England), Greece, Finland, and Slovakia, which illustrate heterogeneity across high-income settings.
We believe this selection of examples offers sufficient breadth and geographic representation to support the narrative’s analytical objectives, while maintaining scientific clarity and readability.
- Breastfeeding is at varying levels – why is this? What are the opinions/attitudes of mothers (and fathers) towards breastfeeding?
Our answer: We thank the reviewer for this pertinent comment. The revised version of Section 2 (Prevalence and Trends of Breastfeeding Globally) now includes a new paragraph addressing the behavioral, social, and cultural factors that explain variations in breastfeeding prevalence across countries. Specifically, we expanded the discussion to include the influence of maternal confidence, partner and family support, workplace and policy barriers, and exposure to commercial milk-formula marketing. We also incorporated evidence on how fathers’ involvement and societal perceptions shape breastfeeding practices, and how cultural ideals regarding modernity and body image contribute to disparities.
- What contributes to such low breastfeeding rates? What are the barriers to breastfeeding in both developing and developed countries?
Our answer: We thank the reviewer for this valuable observation. We have now expanded Section 2 (Prevalence and Trends of Breastfeeding Globally) to include a new paragraph detailing key barriers contributing to low breastfeeding rates across both high- and low-income settings. The revised text addresses structural, sociocultural, and systemic determinants, including inadequate maternity protection, workplace constraints, formula marketing, social stigma, and limited community-based support.
I believe that the above comments will enrich the topic addressed by the authors, and obtaining answers to these questions will help build strategies to raise awareness of the importance of breastfeeding.
Our answer: We sincerely thank once again to Reviewer #2 for the thoughtful and constructive comments. We have considered all comments. We believe they have contributed to strengthen our manuscript and we hope they reflect the above mentioned contributions.